# Retention of lumpy skin disease virus in *Stomoxys* spp (*Stomoxys calcitrans*, *Stomoxys sitiens*, *Stomoxys indica*) following intrathoracic inoculation, Diptera: Muscidae

Arman Issimov[1]*, David B. Taylor[2], Malik Shalmenov[3], Birzhan Nurgaliyev[3], Izimgali Zhubantayev[3], Nurzhan Abekeshev[3], Kaissar Kushaliyev[3], Abzal Kereyev[3], Lespek Kutumbetov[4], Assylbek Zhanabayev[5], Yasmin Zhakiyanova[5], Peter J. White[1]

1 Sydney School of Veterinary Science, Faculty of Science, University of Sydney, Sydney, Australia,
2 Agroecosystems Management Research Unit, USDA-ARS, Lincoln, Nebraska, United States of America,
3 Department of Veterinary Medicine, Zhangir Khan West Kazakhstan Agrarian – Technical University, Oral, Kazakhstan, 4 RGE "Research Institute for Biological Safety Problems" Committee of Science, The Ministry of Education and Science of the Republic of Kazakhstan, Nur-Sultan, Kazakhstan, 5 Department of Veterinary Medicine, Saken Seifullin Kazakh Agrotechnical University, Astana, Kazakhstan

* aiss0820@uni.sydney.edu.au, issimovarman@gmail.com

**Data Availability Statement:** Data are available at doi:10.5063/F1PN9408.

## Abstract

Lumpy skin disease (LSD) is an emerging disease of cattle in Kazakhstan and the means of transmission remains uncertain. In the current study, retention of Lumpy Skin Disease Virus (LSDV) by three *Stomoxys* species following intrathoracic inoculation was demonstrated under laboratory conditions. A virulent LSDV strain was injected into the thorax of flies to bypass the midgut barrier. The fate of the pathogen in the hemolymph of the flies was examined using PCR and virus isolation tests. LSDV was isolated from all three *Stomoxys* species up to 24h post inoculation while virus DNA was detectable up to 7d post inoculation.

## Introduction

Lumpy Skin Disease (caused by LSDV, double-stranded DNA virus, family *Poxviridae*) is an economically devastating disease of cattle [1]. It is a highly contagious disease mainly characterized by multiple skin lesions, fever, enlargement of superficial lymph nodes, excessive salivation, lacrimation and nasal discharge as well as oedema and swelling of the limbs [2, 3]. The World Organization for Animal Health (OIE) classified LSD as a notifiable disease due to its significant economic impact [4]. In addition, it has a detrimental effect on animal production. Economic loss results from a sharp decline in milk yield, milk quality, hide damage, body weight reduction, abortion, infertility and in some cases death of the animal [5]. Morbidity rates vary significantly among LSD outbreaks reaching up to 100%, however low (less than 5%) to moderate rates are more common (20%) [4, 6, 7].

Stable flies (*Stomoxys* spp) have long been considered major pests of livestock in Kazakhstan and are capable of transmitting pathogens among animals. [8]. High infestations of these

**Funding:** Issimov A. This study was funded by "Bolashak" international scholarship program as well as by research project # AP05135323 of the Science Committee/ Ministry of Education and Science/Republic of Kazakhstan. The funders had no role in study design, data collection and analysis, decision to publish, or preparation of the manuscript.

**Competing interests:** The authors have declared that no competing interests exist.

flies are observed in many regions of Kazakhstan with adult *Stomoxys* flies emerging in large numbers following rainfall events in the later months of spring.

To date, LSDV transmission routes and possible insect vectors remain uncertain; however, in controlled studies, mechanical transmission of LSDV was successfully demonstrated using *Stomoxys* species [9, 10] and *Aedes aegypti* mosquitoes [11]. In these studies, the presence of LSDV in *Stomoxys* flies was recorded for a short time, 0–48, hours post feeding on infected blood. LSDV was detected and subsequently isolated from engorged *Stomoxys spp* as well as from their proboscises. However, the question of LSDV viability in *Stomoxys spp* remains to be answered. Thus, the experiments reported here were designed to define the duration of LSDV retention in three *Stomoxys* species following intrathoracic inoculation as well as virus potential to replicate after bypassing a midgut barrier.

## Materials and methods

### Virus strain

In this experiment, LSDV isolate Kubash/KAZ/16 recovered from an infected cow during an outbreak of LSD in Atyrau, Kazakhstan in 2016 were used [12]. Virus was grown on primary lamb testis (LT) and with three in vivo passages through calves. Primary LT cell cultures were derived from pre-pubertal lambs according to Plowright and Ferris [13]. Cell cultures exhibiting90% cytopathic effects (CPE), were freeze-thawed three times, and then centrifuged at 2000 g for 20 min to remove cell debris and stored at -70˚C until required.

### Insect colony

LSDV negative *Stomoxys* spp colonies used in this study were provided by the Entomology section of the Research Institute for Biological Safety Problems (RIBSP). They were maintained in an environmental chamber at 25˚C and 80% humidity, and adults were fed heparinized bovine blood soaked into cotton pads twice daily, 9 a.m. and 5 p.m.

### Design of experiment

As part of an ongoing study of the role of hematophagous flies in the transmission of LSDV in cattle in South Kazakhstan, we utilized three laboratory reared *Stomoxys* species.

Intrathoracic inoculation of LSDV in adult *Stomoxys* spp was conducted according to the protocol published by Rochon, Baker [14] and Perez De Leon and Tabachnick [15]. Prior to experiment commencement, approximately 200 five day old *Stomoxys* flies of each species *(Stomoxys calcitrans, Stomoxys sitiens, Stomoxys indica)* were aspirated into separate plastic cages, anaesthetized using $CO_2$, placed onto a light ice board, and fixed using entomological forceps. Individual flies were inoculated in the dorsal area of the thorax at the junction of the scutum and the scutellum. Virus suspension, 6μl, (Kubash/KAZ/16, $10^6$ $TCID_{50}$/ml) was administered using 100μl Bevel tip 1710N Series Hamilton Gastight microliter syringe (USA). A 26-gauge sterile removable needle was used and changed for each fly (Supelco Inc, Bellefonte, PA). Following virus inoculation, flies were kept in separate cages and maintained at room temperature (24˚C and 70% humidity).

*Stomoxys* flies were tested for the presence of LSDV nucleic acid by gel-based PCR and virus isolation (VI) at 0, 6h, 12h, 24h, 2d, 3d, 4d, 5d, 6d, 7d, 8d, 9d, 10d post intrathoracic inoculation (pii). Pools of ten flies each of each *Stomoxys* species inoculated as described above were collected and tested by PCR and VI immediately following virus administration (0hr). Prior to running tests, pools were rinsed with PBS solution to eliminate contamination. The remaining inoculated *Stomoxys* flies were placed in separate cages according to species and

allowed to recover for 6 h. For sampling purposes, *Stomoxys* spp were aspirated from the cages with a vacuum, anaesthetized in a 5% $CO_2$ gas chamber and transferred to a centrifuge tube and stored at -70°C until tested. Freeze–thawed flies were then washed three times in PBS and rinsed using distilled water to eliminate surface contamination, homogenized in 300μl Hanks solution, spun at 1500 g for 10 min and the supernatant was used for LSDV detection and isolation.

Additionally, three control groups of *Stomoxys* spp *(Stomoxys calsitrans*, *Stomoxys sitiens*, *Stomoxys indica*; 20 individuals of each species) were inoculated by the method previously described and tested using PCR 2 hours after inoculation to determine the total number of infected among inoculated flies as well as to determine the efficacy of the intrathoracic inoculation technique in use.

## Polymerase chain reaction

Virus amplification test was conducted utilizing protocol reported by Tuppurainen, Venter [16]. For DNA extraction, a QIAamp DNA Kit (QIAGEN, USA) was used according to the manufacturer's instructions.

For PCR assay, the forward `5′-TCC-GAGCTC-TTT-CCT-GAT-TTT-TCT-TAC-TAT-3′` and reverse `5′-TAT-GGT-ACC-TAA-ATT-ATA-TACGTA-AAT-AAC-3′` primers were utilized to produce 192 bp of amplified nucleotide reactions. [17]. For DNA amplification in a Thermal Cycler (Eppendorf Mastercycler) following parameters were adjusted: 95°C for 2 min, 95°C for 45 s, 50°C for 50 s, 72°C for 1 min (34 cycles), and 72°C for 2 min. The PCR products obtained were loaded in 1.5% agarose gel electrophoresis in the presence of *Ethidium bromide*, and the results visualized using Bio-Imaging Systems MiniBIS Pro (Israel).

## Virus isolation

Virus isolation was performed according to standard operational procedures (SOP) of the Virology section of the RIBSP, based on OIE [18] guidance. In brief, 10 μl of supernatant were inoculated on to lamb testes cells in 25 $cm^2$ cell culture flasks and incubated at 37°C for 1 h. Following incubation, culture media was washed with Phosphate Buffered Saline (PBS) three times and overlaid with Glasgow's Minimal Essential Medium containing 0.1% penicillin, 0.2% gentamycin and 2% foetal calf serum. The cell monolayer was monitored daily for characteristic CPE. In the case no CPE was observed, the cell culture was freeze–thawed three times and the two or three blind passages were conducted. The culture media used were stored at −70°C until needed. Cell culture flasks exhibiting CPE were tested with gel-based PCR to confirm that CPE was caused by LSDV.

## Analysis

PCR and virus isolation data were analyzed with logistic ANOVA (Proc Glimmix, SAS 9.4). Time post intrathoracic inoculation (pii) was considered a continuous independent variable and species was considered a class independent variable. The interaction between species and pii was included in the model as well.

## Results

Retention of PCR detectable viral DNA did not vary among the three species of *Stomoxys* examined in this study (F = 1.05, df = 2, 33, P = 0.36 for species and F = 0.45, df = 2, 33, P = 0.64 for species*pii interaction). Viral DNA was detectable in 84% of the inoculated flies

immediately following inoculation. Detectability of viral DNA with PCR decreased at the rate of 0.61 logit per day post inoculation (F = 105.17, df = 1, 37, P<0.0001; Fig 1a).

Isolation of infective LSDV did not differ among the three species of *Stomoxys* included in this study (F = 1.65, df = 2, 33, P = 0.21 for species and F = 1.87, df = 2, 33, P = 0.17 for the species*pii interaction. The rate of successful isolation of infective LSDV from *Stomoxys* species decreased by 2.66 logit per day after inoculation (F = 24.77, df = 1, 37, P<0.0001, Fig 1b).

### Demonstration of LSDV retention in *Stomoxys calcitrans*

Virus retention following intrathoracic inoculation procedures resulted in infection in 95% of inoculated flies (Table 1). Pool samples collected for PCR tested positive at time intervals 0, 6h, 12h, 1d, 2d, 3d, 4d, 5d, 6d and 7 days post intrathoracic inoculation (pii) whereas samples tested for virus isolation were LSDV positive with characteristic CPE from time interval 0 up to 24h pii (Table 2).

### Demonstration of LSDV retention in *Stomoxys sitiens*

LSDV DNA retention was detected in 90% of individually infected flies (Table 1). PCR assay was positive for the presence of LSDV in pools samples taken in time intervals 0, 6h, 12h, 1d, 2d, 3d, 4d, 5d and 6 days pii (Table 2). Virus isolation was carried out in cell culture, and CPE characteristic to LSDV was observed in time intervals 0, 6h, 12h pii. Additionally, following the second blind passage virus was recovered from pool samples collected on day 1 pii (Table 2).

### Demonstration of LSDV retention in *Stomoxys indica*

An infection rate composed 95% among intrathoracically infected individuals (Table 1). Viral nucleic acid was detected in time intervals 0, 6h, 12h, 1d, 2d, 3d, 4d, 5d, 6d and 7 days pii (Table 2). CPE similar to LSDV was seen in time intervals 0, 6h, 12h pii while on day 1 pii virus was isolated following the second blind passage in the (LT) cell culture (Table 2).

### Discussion

In this study, the intrathoracic injection of three *Stomoxys* species *(Stomoxys calcitrans, Stomoxys sitiens, Stomoxys indica)* was demonstrated to establish the viability of LSDV in the haemolymph of flies. When LSDV was administered into the hemocoel of adult *Stomoxys* flies, the virus titre decreased over time (Table 2), consistent with the findings reported by Rochon, Baker [14] where Porcine Reproductive and Respiratory Syndrome Virus was detectable using VI up to 24 hours post inoculation. In the present study, the replication of LSDV was not observed in insect tissue which is in agreement with reports published by Chihota, Renniet [19] and Issimov, Kutumbetov [9]. However, the haemolymph does not appear to be inimical to the virus as the alimentary track as demonstrated by relatively prolonged viral DNA presence in intrathoracically inoculated flies. This observation was similar to those described by Schurrer, Dee [20] and Rochon, Baker [14]. All three *Stomoxys* species *(Stomoxys calsitrans, Stomoxys sitiens, Stomoxys indica)* were capable of virus retention after intrathoracic inoculation either in groups (pools) or in individuals. Moreover, viable LSDV was recovered from *Stomoxys spp* up to 48h following virus injection, whereas viral nucleic acid was detected up to day 7 post intrathoracic inoculation.

In a previous study of Porcine reproductive and respiratory syndrome virus retention in house flies, the factors found to influence pathogen stability following *per os* acquisition have been explored by Schurrer [20] and it was found that time and ambient environment have a

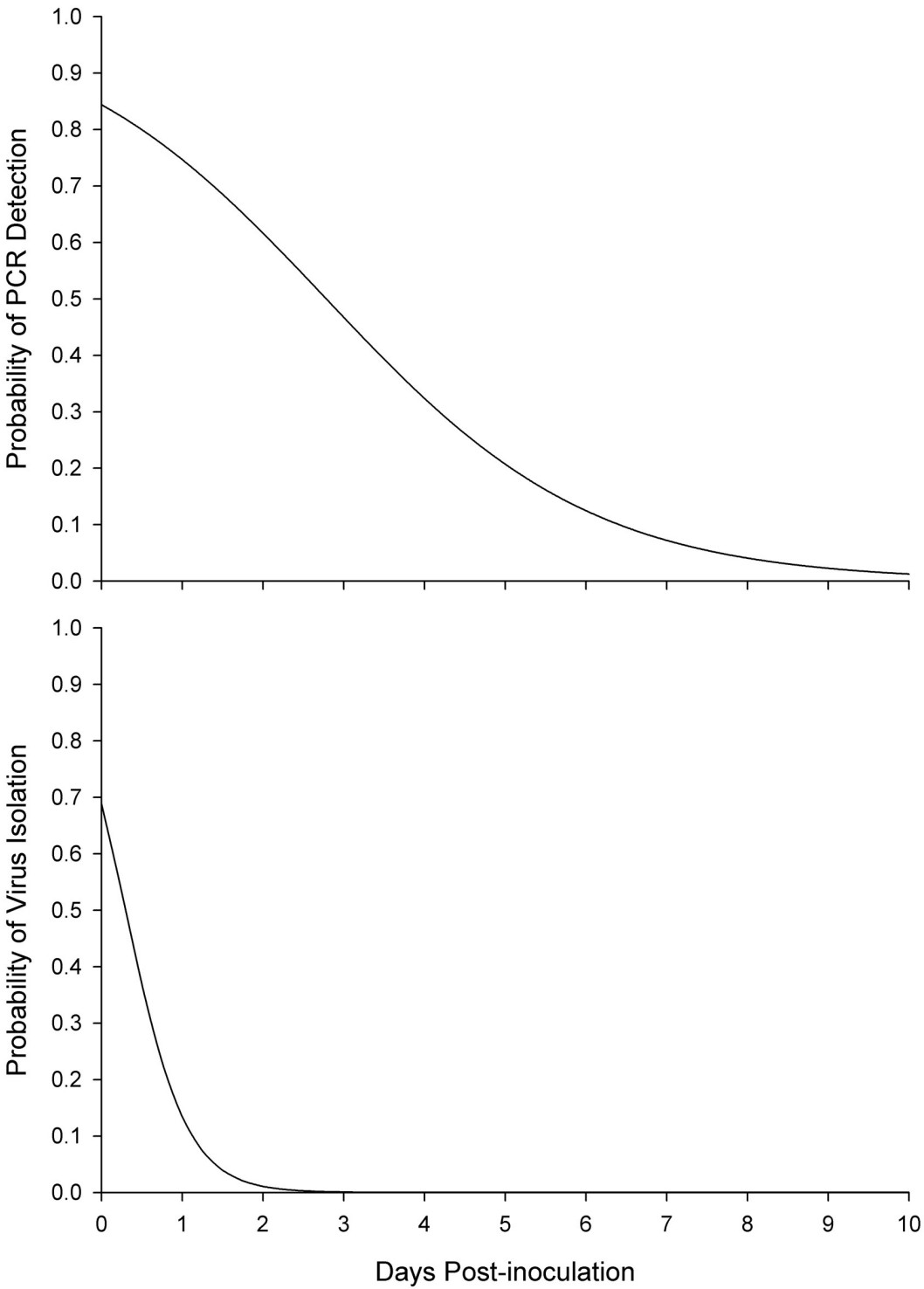

**Fig 1.** **a**—Changes in the concentration of detectable LSDV DNA by PCR at various times postinoculation. **b**—Changes in the titre of infective LSDV by VI at various times postinoculation.

**Table 1. Summary obtained from individual samples of *Stomoxys* species tested 2 hours after inoculation.**

| *Stomoxys* species | Cohort size/individuals inoculated | PCR positive | % infected |
|---|---|---|---|
| *Stomoxys calcitrans* | 20 | 19 | 95 |
| *Stomoxys sitiens* | 20 | 18 | 90 |
| *Stomoxys indica* | 20 | 19 | 95 |

**Table 2. PCR and virus isolation results of *Stomoxys spp* at different time intervals following intrathoracic inoculation with LSDV.**

| Time/day Post inoculation | *Stomoxys calcitrans* | | | *Stomoxys sitiens* | | | *Stomoxys indica* | | |
|---|---|---|---|---|---|---|---|---|---|
| | PCR (no positive/no tested)* | Virus isolation (no positive/no tested)* | Virus isolation ($TCID_{50}$/mL), mean | PCR(no positive/no tested)* | Virus isolation (no positive/no tested)* | Virus isolation ($TCID_{50}$/mL), mean | PCR (no positive/no tested)* | Virus isolation (no positive/no tested)* | Virus isolation ($TCID_{50}$/mL), mean |
| 0h | 10/10 | 7/10 | 4.5 | 9/10 | 9/10 | 4.1 | 10/10 | 7/10 | 3.9 |
| 6h | 8/10 | 4/10 | 3.2 | 9/10 | 5/10 | 2.8 | 8/10 | 6/10 | 2.5 |
| 12h | 7/10 | 2/10 | 2.0 | 8/10 | 4/10 | 1.3 | 6/10 | 3/10 | 1.3 |
| 1 | 7/10 | 2/10 | 1.1 | 6/10 | 1**/10 | - | 5/10 | 1**/10 | - |
| 2 | 7/10 | 1/10 | - | 6/10 | 0/10 | - | 5/10 | 0/10 | - |
| 3 | 6/10 | 0/10 | - | 5/10 | 0/10 | - | 4/10 | 0/10 | - |
| 4 | 5/10 | 0/10 | - | 4/10 | 0/10 | - | 3/10 | 0/10 | - |
| 5 | 3/10 | 0/10 | - | 3/10 | 0/10 | - | 3/10 | 0/10 | - |
| 6 | 1/10 | 0/10 | - | 1/10 | 0/10 | - | 2/10 | 0/10 | - |
| 7 | 1/10 | 0/10 | - | 0/10 | 0/10 | - | 1/10 | 0/10 | - |
| 8 | 0/10 | 0/10 | - | 0/10 | 0/10 | - | 0/10 | 0/10 | - |
| 9 | 0/10 | 0/10 | - | 0/10 | 0/10 | - | 0/10 | 0/10 | - |
| 10 | 0/10 | 0/10 | - | 0/10 | 0/10 | - | 0/10 | 0/10 | - |

*- number of positive and number of tested pools.

**- CPE after second blind passage in (LT) cell culture.

detrimental effect resulting in virus concentration decrease over time in the midgut environment.

In blood-feeding insects, the midgut barrier is a significant line of defense against infection as most of the pathogens are ingested orally via contaminated bloodmeal [21]. The midgut epithelial cells are refractory to infection, and their immunity to viruses is genetically controlled [22]. In addition to this, nonimmune and refractory *individuals* can arise within the same cluster of insects [23]. Prior research has indicated that previously refractory insects may become competent vectors, in the event that a pathogen overcomes the midgut barrier [15]. Studies on the role of *Culicoides* in the transmission of bluetongue virus (BTV) reviled that a colony of intrathoracically BTV infected midges were able to transmit the virus in their saliva. This, in turn, indicates a breach of salivary gland barrier and its susceptibility to infection [23]. Similarly, the integrity of the midgut barrier can also be affected by microfilariae infection and high doses of B. thuringiensis [24, 25]. These findings demonstrate the likelihood that some individuals of insect populations might become permissive enough to allow a new dynamic of the vector—pathogen system [26].

## Conclusion

In conclusion, the outcomes obtained in this study illustrate the incompetence of *Stomoxys* species (*Stomoxys calsitrans*, *Stomoxys sitiens*, *Stomoxys indica*) to serve as biological vectors of

LSDV however further experiments with intrathoracically inoculated flies are of importance to determine if there is a salivary gland barrier for LSDV in the *Stomoxys* fly.

## Supporting information

**S1 Fig.**
(ZIP)

**S1 File. Summary of test results obtained from full model GLMM analyses.**
(ZIP)

## Acknowledgments

The author is sincerely thankful to Mr Aslan Kerembaev and Ms Raihan Nissanova for technical assistance provided.

## Author Contributions

**Conceptualization:** Arman Issimov, Izimgali Zhubantayev, Nurzhan Abekeshev, Abzal Kereyev, Yasmin Zhakiyanova.

**Data curation:** Birzhan Nurgaliyev, Nurzhan Abekeshev, Abzal Kereyev, Lespek Kutumbetov.

**Formal analysis:** Kaissar Kushaliyev, Abzal Kereyev, Assylbek Zhanabayev, Yasmin Zhakiyanova.

**Funding acquisition:** Izimgali Zhubantayev.

**Investigation:** Arman Issimov, Izimgali Zhubantayev, Assylbek Zhanabayev, Yasmin Zhakiyanova.

**Methodology:** Arman Issimov, David B. Taylor, Lespek Kutumbetov.

**Resources:** Malik Shalmenov, Nurzhan Abekeshev.

**Software:** David B. Taylor.

**Supervision:** David B. Taylor, Peter J. White.

**Validation:** Peter J. White.

**Visualization:** Kaissar Kushaliyev, Peter J. White.

**Writing – original draft:** Arman Issimov, Peter J. White.

**Writing – review & editing:** Arman Issimov, David B. Taylor, Peter J. White.

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
