## [Decision Letter · Decision Letter 0]

20 Oct 2020

PONE-D-20-24870

Retention of Lumpy Skin Disease Virus in Stomoxys Spp (Stomoxys Calsitrans, Stomoxys Sitiens, Stomoxys Indica) following intrathoracic inoculation, Diptera: Muscidae.

PLOS ONE

Dear Dr. Issimov,

Thank you for submitting your manuscript to PLOS ONE. After careful consideration, we feel that it has merit but does not fully meet PLOS ONE’s publication criteria as it currently stands. Therefore, we invite you to submit a revised version of the manuscript that addresses the points raised during the review process.

You will notice that the comments from reviewer 1 and reviewer 2 are identical, which is because these two reviewers collaborated to review this manuscript. The reviewers and the editor felt that the manuscript needs to be improved by providing higher quality and more sufficient details of experiments, statistics, other analyses, and data presentation. More critically, reviewer 3 felt that the data is weak in supporting the conclusion. Stronger data and evidence are needed to meet the publication criteria.

If you believe that you can address the reviewer's concerns, please submit your revised manuscript by12/10/2020. If you will need more time than this to complete your revisions, please reply to this message or contact the journal office at plosone@plos.org. Please include the following items when submitting your revised manuscript:

We look forward to receiving your revised manuscript.

Kind regards,

Zhilong Yang

Academic Editor

PLOS ONE

Journal Requirements:

2. We noticed you have some minor occurrence of overlapping text with the following previous publication, which needs to be addressed:

https://www.mdpi.com/2076-2615/10/3/477/htm

In your revision ensure you cite all your sources (including your own works), and quote or rephrase any duplicated text outside the methods section. Further consideration is dependent on these concerns being addressed.

Reviewers' comments:

Reviewer's Responses to Questions

**Comments to the Author**

1. Is the manuscript technically sound, and do the data support the conclusions?

Reviewer #1: Yes

Reviewer #2: Yes

Reviewer #3: No

2. Has the statistical analysis been performed appropriately and rigorously? 

Reviewer #1: N/A

Reviewer #2: N/A

Reviewer #3: No

3. Have the authors made all data underlying the findings in their manuscript fully available?

Reviewer #1: Yes

Reviewer #2: Yes

Reviewer #3: Yes

4. Is the manuscript presented in an intelligible fashion and written in standard English?

Reviewer #1: No

Reviewer #2: No

Reviewer #3: No

5. Review Comments to the Author

Reviewer #1: The authors describe intrathoracic inoculation of three different Stomoxys species with LSDV, to determine retention of LSDV in the hemolymph of these insects. Therefore, they used gel-based PCR assay and virus isolation at different time points after inoculation.

General:

• Be consistent during the whole manuscript by spelling Stomoxys calcitrans, Stomoxys

• sitiens, Stomoxys indica.

• English language: minor improvement necessary

Introduction:

• Could be more references for clinical signs, e.g. Babiuk et al., which is cited later

• Mechanical transmission of LSDV was also demonstrated by Chihota et al. in 2001 and 2003

• L. 43: better “these studies” than “their study”

• L. 45: Stomoxys in italic

• Define the aims of the study in more detail, e.g. why intrathoracic inoculation instead of spiked blood meals? Why is it important to bypass the midgut barrier in this study concept?

Material & Methods:

• L 72: 106 TCID50 per ml??

• Three times passaged in vitro using calves – please explain this

• After how many hours/days were the individual control flies tested?

• Was a pool of flies always 10 flies/pool or only at 0h?

• Pools of flies were homogenized in 300 µl Hanks solution, but 1 mL of suspension was used for virus isolation – how was the volume increased?

Results:

• Again, when were the individual control flies tested?

• L 119: Pool samples

• L 131: at which stage?

Discussion:

• Decrease of titer is in contrast to Rochon, Baker, who, due to the reference cited, examined Porcine Reproductive and Respiratory Syndrome Virus. This should be clarified in the text.

• Use of “in this study” is misleading when describing the present study

• Reference for impact and description of the mentioned impact of the alimentary tract on LSDV is missing

• Again: Schurrer, Dee also examined porcine reproductive and respiratory syndrome virus, this should be mentioned to be clear

Reviewer #2: The authors describe intrathoracic inoculation of three different Stomoxys species with LSDV, to determine retention of LSDV in the hemolymph of these insects. Therefore, they used gel-based PCR assay and virus isolation at different time points after inoculation.

General:

• Be consistent during the whole manuscript by spelling Stomoxys calcitrans, Stomoxys sitiens, Stomoxys indica.

• English language: minor improvement necessary

Introduction:

• Could be more references for clinical signs, e.g. Babiuk et al., which is cited later

• Mechanical transmission of LSDV was also demonstrated by Chihota et al. in 2001 and 2003

• L. 43: better “these studies” than “their study”

• L. 45: Stomoxys in italic

• Define the aims of the study in more detail, e.g. why intrathoracic inoculation instead of spiked blood meals? Why is it important to bypass the midgut barrier in this study concept?

Material & Methods:

• L 72: 106 TCID50 per ml??

• Three times passaged in vitro using calves – please explain this

• After how many hours/days were the individual control flies tested?

• Was a pool of flies always 10 flies/pool or only at 0h?

• Pools of flies were homogenized in 300 µl Hanks solution, but 1 mL of suspension was used for virus isolation – how was the volume increased?

Results:

• Again, when were the individual control flies tested?

• L 119: Pool samples

• L 131: at which stage?

Discussion:

• Decrease of titer is in contrast to Rochon, Baker, who, due to the reference cited, examined Porcine Reproductive and Respiratory Syndrome Virus. This should be clarified in the text.

• Use of “in this study” is misleading when describing the present study

• Reference for impact and description of the mentioned impact of the alimentary tract on LSDV is missing

• Again: Schurrer, Dee also examined porcine reproductive and respiratory syndrome virus, this should be mentioned to be clear

Reviewer #3: Lumpy skin disease (LSD) is an economically important transboundary disease of cattle, which is mainly transmitted by arthropod vectors. Cattle are the natural hosts for LSDV, and Asian water buffaloes can also be naturally infected with LSDV. LSDV is highly host restricted. LSDV can be transmitted by herd direction transmission or mechanically transmitted by different blood-feeding vector. In this manuscript, Arman et al evaluated the susceptibility of Stomoxys spp by intrathoracic inoculation with LSDV. All three stomoxys species were intrathoracic inoculated with LSDV. The results showed that LSDV was isolated from all three stomoxys species immediately and continued to 24h post-inoculation. The PCR positive results can be detected up to 7d post-inoculation. Overall, the data looked extremely weak. It is difficult to justify the conclusions with such weak data.

Specific comments:

1) Although LSDV was isolated from all three stomoxys species, peak virus titer was found at 0h and quickly decreased within 24h, indicating the LSDV failed to replicate in stomoxys species. The author previous published paper demonstrated the transmission of LSDV by three stomoxys species under laboratory condition. These species may not transmit LSDV on natural condition.

2) In addition to virus retention, the virus transmission by these stomyxys species, which is more interesting aspect should be included in this manuscript. Unfortunately, the virus transmission was published by author.

6. PLOS authors have the option to publish the peer review history of their article (what does this mean?). If published, this will include your full peer review and any attached files.

Reviewer #1: No

Reviewer #2: No

Reviewer #3: No

---

## [Author Response · Author response to Decision Letter 0]

26 Nov 2020

To make a manuscript more saturated, we added statistical analysis using SAS software.

---

## [Editor Report · Decision Letter 1]

1 Dec 2020

Retention of Lumpy Skin Disease Virus in Stomoxys Spp (Stomoxys Calsitrans, Stomoxys Sitiens, Stomoxys Indica) following intrathoracic inoculation, Diptera: Muscidae.

PONE-D-20-24870R1

Dear Dr. Issimov,

We’re pleased to inform you that your manuscript has been judged scientifically suitable for publication and will be formally accepted for publication once it meets all outstanding technical requirements.

Kind regards,

Zhilong Yang

Academic Editor

PLOS ONE
---

## [Editor Report · Acceptance letter]

20 Jan 2021

PONE-D-20-24870R1 

Retention of Lumpy Skin Disease Virus in *Stomoxys* Spp (*Stomoxys calcitrans, Stomoxys sitiens, Stomoxys indica*) following intrathoracic inoculation, Diptera: Muscidae. 

Dear Dr. Issimov:

I'm pleased to inform you that your manuscript has been deemed suitable for publication in PLOS ONE. Congratulations! Your manuscript is now with our production department. 

Kind regards, 

on behalf of

Dr. Zhilong Yang 

Academic Editor

PLOS ONE